

# Two different mechanisms cause severe hot droughts in the western Amazon

Katharina Baier[1], Lucie Bakels[1], and Andreas Stohl[1]

[1]Department of Meteorology and Geophysics, University of Vienna, Vienna, Austria

**Correspondence:** Katharina Baier (k.baier@univie.ac.at)

**Abstract.** In the last few decades, the Amazon basin has experienced several periods of simultaneous extreme drought and heat. It is crucial to gain a better causal understanding of these compound events, which can severely damage the rain forest ecosystem. Here, we study the role of atmospheric mass, moisture and heat transport to the western Amazon, an area where pristine tropical rain forests can still be found. To investigate how anomalous the atmospheric transport during extreme events

is compared to the climatological mean, we use a Lagrangian reanalysis dataset created with the particle dispersion model FLEXPART driven with meteorological input data from the ERA5 reanalysis. For the period 1979-2021, air masses over the western Amazon are selected every three hours and traced 10 days backward in time. We compared the overall transport climatology for the end of the dry season (August-September) with the transport for 21 extreme compound events that were selected for their extremity in terms of high air temperature and low soil water content. We find that extreme events over the

western Amazon that happen during El Niño events have very different causes from those that take place under La Niña or neutral ENSO (El Niño Southern Oscillation) conditions. For the five extreme events that occurred under El Niño conditions, around 50 % of the air, as compared to 30 % for the climatology, is located over the tropical Atlantic Ocean 10 days prior to arrival. Air that is already anomalously dry and warm is transported from this region towards the western Amazon, highlighting the role of long-range transport for these extremes. The other 16 extreme events occurred under La Niña or neutral ENSO

conditions and show similar transport patterns to the overall transport climatology until 3 days prior to arrival. During the last 3 days, however, air is preferentially transported over the southern Amazon, where already dry soil conditions cause a malfunctioning of moisture recycling, resulting in the propagation of extreme conditions to the western Amazon. Air arriving in the western Amazon during these events is travelling over areas heavily affected by deforestation. Therefore, we expect that landuse changes over the southern Amazon have a stronger impact on compound drought and heat extremes in the western

Amazon under La Niña or neutral ENSO conditions than during El Niño events.

## 1  Introduction

The Earth's largest rain forest ecosystem, the Amazon, holds significant importance for our climate, as it plays a crucial role in both the water and carbon cycle. For instance, it absorbed between 0.42 and 0.65 Pg of carbon annually between 1990 and 2007 (Pan et al., 2011). However, droughts and heat waves can severely damage the rain forest, as tree mortality increases under dry

conditions (Phillips et al., 2010; Doughty et al., 2015). As a consequence, the forest takes up less carbon (Potter et al., 2011) or





can even lose carbon to the atmosphere (e.g. Phillips et al., 2009; Reichstein et al., 2013). Furthermore, these conditions favor the occurrence of fires Aragão et al., 2018. Heat waves in the Amazon are also associated with a higher mortality rate of the human population and negative socioeconomic impacts (Silveira et al., 2023).

Although the Amazon region is usually wet, as it is located within the Tropics, it experienced multiple droughts and heat waves in the last few decades. For example the drought in 2010 caused huge changes in the carbon cycle (Potter et al., 2011). Other extreme droughts happened, e.g. in 2005 and 2015 (e.g. Marengo et al., 2016; Costa et al., 2022). However, the individual extreme events differ both in their spatial extent and duration. For instance, Lewis et al. (2011) showed that the drought in 2010 extended over wider areas compared to the one in 2005. More recently, extremes occurred in 2020 (Marengo et al., 2022), especially in central South America, and in 2023 an extreme drought affected the Negro river region (Rodrigues, 2023). Several droughts occurred together with extreme heat waves (Marengo et al., 2022; Costa et al., 2022).

While the Amazon is a well-studied region, only little is known about the mechanisms causing droughts and heat waves there. So far such events have been mainly linked to a northward shift of the Inter-Tropical Convergence Zone (ITCZ) (e.g. Geirinhas et al., 2018), a reduction in moisture import fluxes (Costa et al., 2022) and anomalies in Atlantic sea surface temperatures (SST) (e.g. Marengo et al., 2011; Lewis et al., 2011). Costa et al. (2022) studied compound events over southeastern Amazonia and found that the reduction in moisture influx is related to the strengthening of the South Atlantic Anticyclone, which is linked to the South American Low Level Jet (SALLJ). The main driver of the 2005 and 2010 droughts was found to be an anomalously warm Atlantic Ocean (e.g. Marengo et al., 2008; 2011). Although several events occurred during positive phases of the El Niño Southern Oscillation (ENSO) (Jiménez-Muñoz et al., 2016), no clear correlation with ENSO exists as other events occurred during neutral or negative ENSO phases.

Lagrangian modeling has become a widely used technique to study the mechanisms causing extreme events (Zschenderlein et al., 2020; Bieli et al., 2015). Stohl and James (2004) introduced a moisture tracking technique to identify sources of evaporation and precipitation, both climatologically on a global scale and for individual events of, e.g. extreme rainfall. Other studies have used Lagrangian modelling to better understand heat waves (e.g. Zschenderlein et al., 2020; Röthlisberger and Papritz, 2023; Baier et al., 2023) by bookkeeping of adiabatic and diabatic heating and cooling of the air along trajectories. Using this information, source regions of the heat and physical mechanisms causing the heating can be identified and the role of atmospheric transport can be clarified.

Drumond et al. (2014) studied the hydrological processes important for the Amazon basin by using the Lagrangian model FLEXPART and the moisture tracking method by Stohl and James (2004). They found that the primary moisture source for the Amazon basin is the tropical Atlantic Ocean. Similar results were found by Leyba et al. (2023) who highlighted the importance of the tropical and south tropical Atlantic Ocean for the Amazon moisture budget. However, these studies only examined the climatological moisture sources and did not investigate extreme events.

Here, we study the role of atmospheric transport for extreme droughts under extreme heat over the western Amazon. We first identify such compound events and then, using a Lagrangian reanalysis data set, trace the air from the western Amazon 10 days backward in time. We compare the atmospheric transport leading to the extreme events to the climatological transport for the period from 1979 to 2021.



## 2 Methods

### 2.1 Extreme events over the western Amazon

We study extreme compound drought and heat events (CDHEs) in a region of pristine rainforest in the western Amazon, defined as our target region $WAM_t$ (Fig. 1b) during August and September (AS) when dry and warm conditions predominate. We also considered June and July (JJ), but did not find any extreme events during that period, thus we only compare to the AS climatology. As the rain forest is damaged most severely when high temperatures occur simultaneously with low soil moisture, we identify the extremes by using a combined threshold approach as described by Hao et al. (2022):

$$Z = \begin{cases} 1, & W < W_0 \text{ and } T > T_0 \\ 0, & \text{others} \end{cases} \tag{1}$$

where $W$ is the daily minimum of the volumetric soil water for the top layer (0-7 cm) and $T$ the daily maximum 2 metre air temperature from ERA5 (Hersbach et al., 2020). We set $W_0$ to the 10th percentile of all $W$ values and and $T_0$ to the 90th percentile of all $T$ values. We apply equation 1 to each $0.5° \times 0.5°$ grid cell in the $WAM_t$ region for JJ and AS months during the period 1979 to 2021, i.e. for around 5000 days in total. A day when these thresholds are exceeded in at least 50 % of the $WAM_t$ grid cells is considered as an extreme day. To define a CDHE event, we apply the additional criterion that at least 3 consecutive days must fulfill our extreme day criterion. This is motivated by the fact that short drought periods may be tolerated by the rain forest, whereas persistent droughts are more damaging. Based on these strict definitions, we find 21 extreme CDHEs in the $WAM_t$ region, with a total duration of 115 days, as listed in Table 1.

### 2.2 Lagrangian reanalysis

We use a Lagrangian Reanalysis (LARA) data set created with the newest version of the Lagrangian dispersion model FLEX-PART (based on Pisso et al., 2019; Bakels et al., 2024). For the meteorological input to FLEXPART we used ERA5 meteorological reanalysis data (Hersbach et al., 2020) with a horizontal resolution of $0.5° \times 0.5°$, 137 model levels and a temporal resolution of one hour. The LARA data set used in this study covers the period from 1979 to 2021, with particle positions and meteorological information reported to output files every hour. A total of 6 million particles of equal mass (each carrying a 6 millionth fraction of the total mass of the atmosphere) are initially distributed in the global atmosphere according to air density as given by the ERA5 data, and traced forward in time. This domain-filling set-up was similar to the one used by Stohl and James (2004). A recent application was reported by Baier et al. (2022).

### 2.3 Trajectory analysis

For each extreme event and for each monthly mean of the climatology, we checked LARA particle positions every three hours, selected all particles residing over our $WAM_t$ target region (Fig. 1b, orange) and traced them backward in time for 10 days. We selected particles over the whole atmospheric column but tracked them separately for four different target vertical layers. The first layer includes all $WAM_t$ particles within the lowest 1 km, representing the boundary layer where particles have the





strongest impact on the surface conditions. The particles between 1 and 2 km are most representative of the low clouds in the WAM$_t$ region (low layer), particles between 2 and 5 km represent the medium cloud cover layer (medium layer), and particles between 5 and 20 km (high layer) are representative of the high cloud cover. In the following we focus on the particles arriving in the boundary layer and the medium layer because the highest layer shows only small differences between the extremes and

the climatology (Fig. A2) and the low cloud layer results are similar to those of the lowest layer (Fig. A1). By comparing the results for the individual layers, we are also able to investigate how differential temperature advection towards these layers affects atmospheric stability.

We investigate the mass transport represented by all selected particles and their source regions by calculating their relative mass contribution to the total atmospheric mass contained in an atmospheric column, including the environmental air not

represented by the selected particles. This is done for specific regions and as a function of backward simulation time, i.e., the time before the particles' selection over the WAM$_t$ region (see Baier et al., 2022 for more details). We focus our analysis on transport from three continental regions: the southern Amazon (SAM), eastern Amazon (EAM) and western Amazon (WAM), where the WAM region includes the target region (WAM$_t$) and is further extended to the north and west. Additionally, we investigate transport from two oceanic regions: the Tropical Atlantic (TAT) and the southern Atlantic Ocean (SAT).

We also tracked several meteorological parameters along the trajectories (e.g. temperature $T$, specific humidity $q$, height $z$ and pressure $p$) and derived other parameters based on those (e.g. potential temperature $\theta$, equivalent potential temperature $\theta_e$). The thermodynamic properties and their changes along the trajectories allow us to diagnose processes such as adiabatic and diabatic heating, as well as moisture uptake and release (Baier et al., 2023). For this purpose, we plot the median meteorological parameters of all selected particles as a function of the backward time. We compare the 21 extreme events to the climatology,

which is presented as an average over the period from 1979 to 2021 over AS.

## 3    Results

### 3.1    Extreme compound drought and heat events (CDHEs)

We identified 21 CDHE cases during the period 1979-2021, of which 20 occurred in the 21st century (Tab. 1). The events had a typical duration of around 5 days but the longest event extended over 20 days, occurring in 2015 at the beginning of a strong

El Niño. In some years (2005, 2010, 2011, 2012 and 2020), we found more than one CDHE case. These events might be linked to each other, as earlier dryness preconditions the soil for the development of a new drought.

We checked the Nino3.4 (N3.4) index for the month of each event (0M) and with a time-lag of +3 months (+3M), as El Niño usually starts in boreal autumn, peaks during boreal winter and persists into boreal spring. We defined all events with N3.4 index values greater than 0.5 K for 0M and +3M as El Niño events (E+). We found that 5 out of 21 CDHE cases happened

under El Niño conditions. This includes the extremely strong El Niño events as in 1997 and 2015. 11 out of 21 events happened during La Niña, with N3.4 index being less than -0.5 K for 0M or +3M, including some triple-dip La Niña events as in 2020 and 2021. The remaining 5 events took place under neutral ENSO conditions. Thus, WAM$_t$ CDHE events occur in all ENSO phases. Based on a previous study (Baier et al., 2022) that showed that droughts over the Amazon region are strongly influenced





by El Niño, we expect that processes causing CDHEs in the $\mathrm{WAM}_t$ are different for El Niño and other events. Therefore, we
distinguish between CDHEs occurring under El Niño conditions (referred to as E+ cases) and all other events (referred to as
E0- cases).

**Table 1.** *Compound drought and heat events (CDHEs) over the western Amazon during the period 1979-2021.* For each event, its starting
date and duration in days, the minimum value of the volumetric soil water in layer 1 (0-7 cm, swvl1), the maximum of the 2 meter air
temperature, the anomaly in sea surface temperature (SSTA) over the southern Atlantic Ocean (SAT), and the Nino3.4 index given by NOAA
(Rayner et al., 2003) for the month of the extreme event (0M) as well as with a time lag of plus three months (+3M), are reported. The last
column (E) shows our classification of the CDHE cases into two groups according to the N3.4 index: E+ and E0-.

|   | DATE | Days | $swvl1_{min}$ (m³/m³) | $T_{max}$ (K) | SAT SSTA (K) | N3.4(0M) | N3.4(+3M) | E |
|---|------|------|-----------------------|---------------|--------------|----------|-----------|---|
| 1 | 1997-09-17 | 5 | 0.3689 | 305.85 | -0.02 | 2.1 | 2.3 | E+ |
| 2 | 2002-09-21 | 4 | 0.3703 | 306.21 | -0.17 | 0.82 | 1.41 | E+ |
| 3 | 2005-08-13 | 3 | 0.3447 | 305.40 | 0.32 | -0.04 | -0.44 | E0- |
| 4 | 2005-08-29 | 4 | 0.3640 | 306.28 | 0.32 | -0.04 | -0.44 | E0- |
| 5 | 2005-09-23 | 4 | 0.3481 | 307.65 | 0.41 | -0.08 | -0.75 | E0- |
| 6 | 2006-09-02 | 3 | 0.3751 | 306.27 | 0.36 | 0.63 | 1.1 | E+ |
| 7 | 2007-08-24 | 3 | 0.3712 | 305.09 | -0.30 | -0.57 | -1.58 | E0- |
| 8 | 2009-09-04 | 7 | 0.3545 | 306.19 | -0.04 | 0.68 | 1.81 | E+ |
| 9 | 2010-08-17 | 3 | 0.3515 | 305.17 | 0.39 | -1.33 | -1.57 | E0- |
| 10 | 2010-08-23 | 6 | 0.3551 | 305.79 | 0.39 | -1.33 | -1.57 | E0- |
| 11 | 2010-09-10 | 5 | 0.3682 | 306.12 | 0.19 | -1.56 | -1.63 | E0- |
| 12 | 2011-08-08 | 5 | 0.3526 | 305.56 | -0.02 | -0.66 | -1.09 | E0- |
| 13 | 2011-08-26 | 6 | 0.3540 | 306.52 | -0.02 | -0.66 | -1.09 | E0- |
| 14 | 2012-08-05 | 5 | 0.3456 | 305.17 | 0.52 | 0.66 | 0.33 | E0- |
| 15 | 2012-08-21 | 4 | 0.3594 | 305.58 | 0.52 | 0.66 | 0.33 | E0- |
| 16 | 2012-09-19 | 4 | 0.3561 | 306.44 | 0.56 | 0.44 | -0.13 | E0- |
| 17 | 2015-09-09 | 20 | 0.3248 | 307.56 | 0.16 | 2.01 | 2.56 | E+ |
| 18 | 2017-08-11 | 9 | 0.3450 | 306.35 | 0.14 | -0.18 | -0.84 | E0- |
| 19 | 2020-08-18 | 3 | 0.3494 | 306.55 | 0.42 | -0.42 | -1.01 | E0- |
| 20 | 2020-09-17 | 5 | 0.3574 | 306.38 | 0.40 | -0.66 | -0.98 | E0- |
| 21 | 2021-08-20 | 7 | 0.3323 | 307.02 | 0.28 | -0.38 | -0.88 | E0- |

## 3.2    Source regions of the transported air

Most of the air arriving in our $\mathrm{WAM}_t$ target region is advected from the Atlantic Ocean by easterly winds (Fig. 1b). We find
that, averaged over the 10 days of transport, around 35 % of the tracked air is located over the extended WAM region (Fig.
2a), while around 7 % and 20 % are located over the southern (SAM) and the eastern Amazon (EAM). Around 10-15 % of



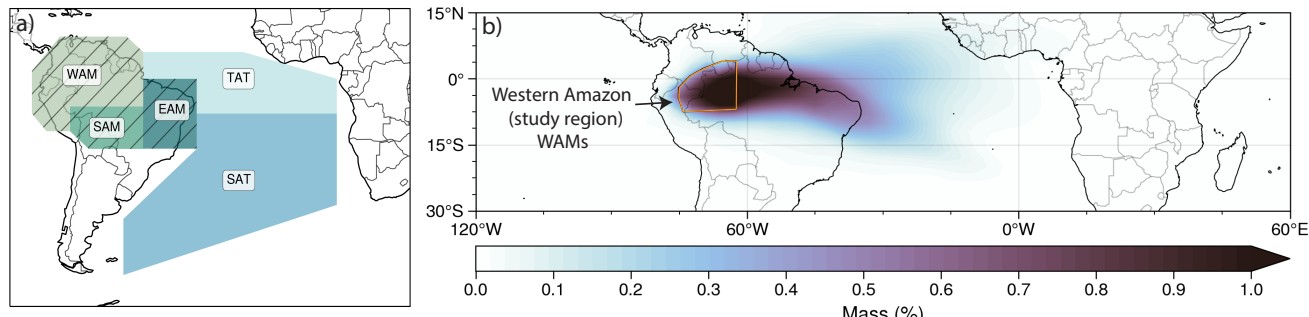

**Figure 1.** *Mass transport climatology for August and September.* (a) Definition of the regions: western Amazon (WAM), southern Amazon (SAM), eastern Amazon (EAM), Tropical Atlantic Ocean (TAT) and southern Atlantic Ocean (SAT) (Iturbide et al., 2020). (b) Climatological mean mass fraction of particles tracked backward from the $WAM_t$ target region (demarcated by the orange line), integrated over the total atmospheric column and for the whole 10 day tracking period.

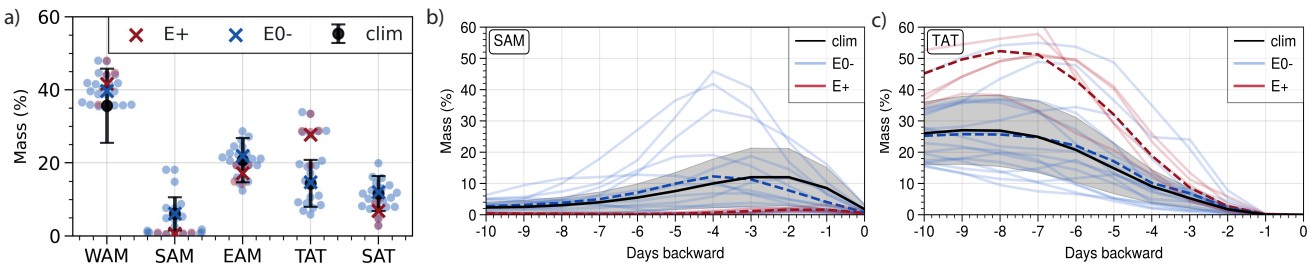

**Figure 2.** *Mass transport for August and September, for the climatology and for the CDHE cases.* (a) Relative source region contributions to the mass transport (%) for each CDHE case and for the climatology, calculated over the whole 10 day tracking period. The blue dots show the values for CDHE cases under neutral or La Niña conditions (E0-) and the red dots show all cases under El Niño conditions (E+), and the crosses represent the respective means over all CDHE cases. The black point represents the mean of the climatology with one standard deviation (whiskers). (b, c,) Evolution of relative mass fraction contributions in % as a function of backward time, for the two lowest target vertical layers combined, for source region SAM (d) and TAT (e). Blue solid lines represent E0- cases, red solid lines represent E+ cases, and the black solid line represents the climatological mean. Shaded grey areas represent one standard deviation for the climatology.

the transported mass is located over the Tropical (TAT) and southern Atlantic Ocean (SAT) each. While all the tracked air is starting over a WAM subregion ($WAM_t$) and thus also spends considerable time there before leaving it, when going further back in time it is crossing mostly the EAM region. The mass transport over the EAM region is reaching its peak, of 40 %, at around day -4 (Fig. A3b). Only little air is transported over the SAM region but this contribution also peaks around day -3 (Fig. 2b). Between about days -5 and -10, we see a strong increase in mass over the Atlantic Ocean. At day -10, around 30 % of the tracked mass is found over the TAT and SAT each (Fig. 2c and Fig. A3c), whereas less than 10 % of the air is still residing





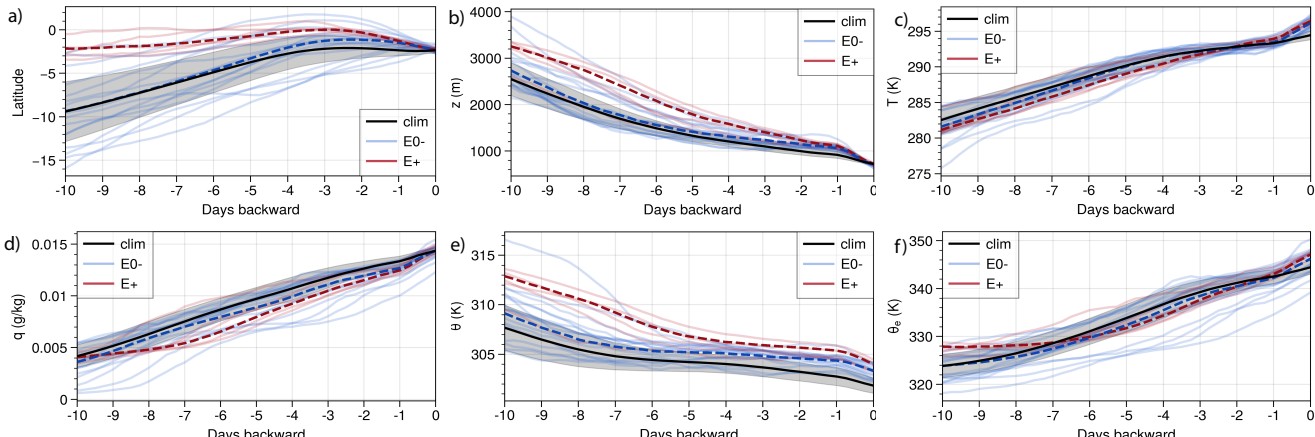

**Figure 3.** *Meteorological parameters as a function of time prior to arrival of particles traced back from the lowest vertical layer (0-1 km) in the WAM$_t$ target region.* Light blue solid lines represent the E0- cases, light red solid lines represent the E+ cases and the black solid line represents the climatological mean. The dashed blue line shows the mean over all E0- cases and the dashed red line shows the mean over all E+ cases. Shaded grey areas represent one standard deviation around the climatological mean. The following parameters are shown: latitude (a), altitude above sea level $z$ (b), air temperature $T$ (c), specific humidity $q$ (d), potential temperature $\theta$ (e) and equivalent potential temperature $\theta_e$ (f).

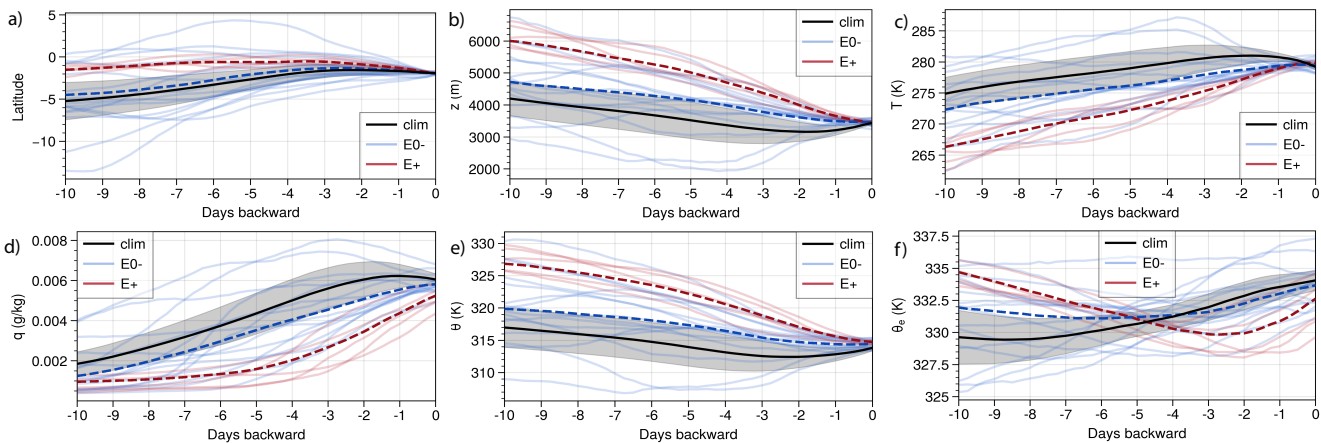

**Figure 4.** *Meteorological parameters as a function of time prior to arrival of particles traced back from the medium layer (2-5 km).* Same as in Figure 3, but for the medium layer.

over the WAM (Fig. A3a). Based on this, we note that oceanic impacts predominantly occur from days -10 to -5, whereas continental impacts are more important during the last five days before arrival.





### 3.3 Extreme events during El Niño (E+)

For the E+ cases, we find a strong oceanic component, with about 25 % of the 10-day travel time spent over the tropical Atlantic Ocean (Fig. 2a). On average, the particles for E+ cases are located at around 2.5 °S 10 days prior to arrival, while the corresponding value for the climatology is around 10 °S (Fig. 3a, 4a). Particularly striking is that the air spends almost no time over the SAM region for the E+ cases. Furthermore, for E+ cases the air originates from higher altitudes than for the climatology (Fig. 3b, 4b). Even the air arriving in the lowest 1 km of the $WAM_t$ originates from mid-tropospheric levels

(mostly over the TAT region), likely air that has previously been lifted by convection over the eastern part of the TAT region or Africa.

Since the E+ cases are strongly influenced by the tropical Atlantic Ocean, we also find systematic differences in the meteorological parameters of the transported air compared to the climatology. The particles traced from the lowest 1 km of the $WAM_t$ target region have higher potential temperature values for the E+ cases than for the climatology throughout the whole tracking

period (Fig. 3e). Ten days prior to arrival, potential temperature, $\theta$, is around 313 K for E+ cases, while it is only around 308 K +/- 2 K for the climatology. This highlights the importance of long-range advection of warm air towards $WAM_t$ for the E+ compound events. While there is slightly less diabatic heating than for the climatology, seen by the slower $\theta_e$ increase compared to the climatology (Fig. 3f), $\theta$ values are still greater by about 3 K for E+ cases than for the climatology when the air arrives in the $WAM_t$ target region. The stronger descent of the air for E+ cases also causes stronger adiabatic heating, resulting

in a stronger increase in temperature than for the climatology (Fig. 3b, c). This is particularly pronounced during the last 24 hours, resulting in a temperature difference of 3 K upon arrival.

The specific humidity at ten days prior to arrival is slightly lower for the E+ cases than for the climatology (Fig. 3d) and it remains at these low levels for three days (-10 to -7), confirming that the air has little contact with the ocean surface. The air also remains drier than for the climatology throughout the 10-day tracking period. Generally, the meteorological parameters

are characteristic for mid-tropospheric air that has been lifted by convection previously within the Tropics. While specific humidity upon arrival is only slightly lower than for the climatology, indicating moistening as the air enters the continent and becomes entrained into the boundary layer, relative humidity is clearly lower than for the climatology because of the higher air temperature.

The moisture deficit for the E+ cases is even more pronounced for the air arriving in the $WAM_t$ target region in the medium

vertical layer (2-5 km) (Fig. 4d). As there is less exchange with the surface than for the lower layer, there is also less moistening over the continent and the air arrives with a substantial moisture deficit compared to the climatology. This inhibits cloud formation in the lower to mid-troposphere for the E+ cases. The consequent reduction in clouds (not shown here) increases the solar irradiation penetrating to the surface, thus further favoring the development of extreme heat and droughts over the $WAM_t$ region. To clarify the role of the soil moisture conditions over continental areas, we look at the soil water anomalies over the

SAM (Fig. 5d) and $WAM_t$ regions (Fig. 5e) before each extreme event. For E+ cases no dry anomaly in soil water content over SAM and no transport from there is found. Even over the $WAM_t$ region itself, dry anomalies in the soil water content start to develop only in the last two days before the event (Fig. 5e), thus confirming our previous finding that long-range transport



from the TAT region is the controlling factor for the E+ events, whereas surface exchange processes over the continent appear to have a very limited influence.

In summary, for E+ cases, advection of warm and dry mid-tropospheric air from the tropical Atlantic Ocean, is the most important driver. The air is transported from the overall warmer Tropics, crossing the South American coast, towards the western Amazon. The air is also transported at overall higher altitudes and fewer of the tracked particles are located within the lowest 1 km (Fig.5a-c) than for the climatology. Therefore, the air is in less contact with the surface than for the climatology, reducing the influence of moisture recycling in the rain forest. This is also accompanied by a deficit in mass transport over

the EAM and SAM regions (Fig. 2a). Therefore, we expect co-occurring droughts over these areas playing a small role for E+ cases. Instead, conditions over the Atlantic Ocean are the major factor controlling the severity of E+ CDHE cases.

### 3.4   Extreme events during La Niña and neutral conditions (E0-)

We find similar mass transport patterns for most E0- CDHE cases. The WAM mass contributions for most E0- CDHE cases is within one standard deviation of the climatology, but usually slightly above the climatological mean, thus indicating somewhat

more stagnant conditions (Fig. 2a). Some E0- cases also have a strong southerly transport component (Fig. 3a, 4a), with multiple events showing mass contributions significantly above the climatological mean over the SAM region (Fig. 2a). The mass contribution from the SAM region for these cases peaks around day -4, reaching values of 20 to 45 % (Fig.2b) for five events in 2005, 2010 and 2012. As more air is transported over the SAM region, we expect a strong link between the southern and western Amazon for these events, as discussed in the following sections.

In contrast to the E+ cases, the particles arriving in the $WAM_t$ target region within the lowest 1 km for E0- cases show overall similar meteorological characteristics to the climatological mean (Fig. 3). They originate from a similar altitude, and specific humidity $q$ (Fig. 3d) and potential temperature $\theta$ (Fig. 3e) are similar to the climatology ten days prior to arrival, with average differences of only around 1.5 K and less than 0.001 g/kg. Still, the E0- cases are slightly warmer and drier than the climatology throughout the whole tracking period. During the last three days before arrival the particles are sinking at a slower

pace for the E0- cases compared to the climatology (Fig. 3b), and $\theta$ values exceed the climatological mean by more than one standard deviation. Within the last day, the air is sinking faster, thus stronger adiabatic heating takes place (Fig. 3c). This is likely caused by higher boundary layer heights (Tab. A1) than for the climatology, which causes stronger entrainment. Overall, there seems to exist a stronger surface-atmosphere coupling over the continent during the last few days of transport, and a tendency for some cases for this to occur over the SAM region.

A few of the E0- cases show clear differences to the climatology throughout the whole 10-day tracking period. While all E+ events are warmer already 10 days prior to arrival, some of the extreme anomalies are actually found for E0- events. This is the case for the following four events: 2011-08-08, 2010-08-17, 2020-08-18 and 2012-08-05, where the latter one is specifically striking with a $\theta$ value of 317 K ten days prior to arrival (Fig. 3e). We find that all four events are the first extreme event occurring in the given year, and all are followed by at least one additional event later in the season. Thus, it seems advection

of warm air is an important driver not only for the E+ cases but also for the first E0- case of a season, which often triggers a subsequent event that is more influenced by the preconditioning of the soil within the Amazon region by the first event.





We found for all E0- cases that the SAT is anomalously warm (Tab. 1). This is consistent with previous studies that high-lighted the role of positive SST anomalies over the Atlantic Ocean for droughts over the Amazon (e.g. Lewis et al., 2011). However, only a few E0- cases, usually the first ones of the year, show also higher potential temperatures at day -10. Thus, we

expect that these events are directly influenced by the warmer Atlantic Ocean. This is specifically striking for the 2012-08-05 event, with a SST anomaly greater than $0.5\,\text{K}$ and with $55\,\%$ of the air located over the SAT region ten days prior to arrival, compared to $33\,\%$ for the climatology. Therefore, advection of anomalously warm air coming from a warmer SAT seems to trigger the first extreme E0- event within one year.

In contrast to the E+ events, we find for most E0- events drier surface conditions in both the WAM as well as the SAM

region already 20 days before the start of the extreme event (Fig. 5). While most events show a small mass fraction over SAM (Fig. 1d), we found a strong SAM component for the following events: 2005-08-05, 2010-08-17 and 2012-08-05 with peaks of around $44\,\%$, $50\,\%$ and $36\,\%$ at day -4. These are accompanied with low $q$ values for the particles arriving in the lowest $1\,\text{km}$ (Fig. 3f). This might be linked to a reduction in evaporation over the southern Amazon (Fig. 3d), due to dry surface conditions (Fig. 5d). All three events are the first ones occurring in that given year. This shows that due to anomalous dry conditions over

SAM, the transported air took up less moisture than usual, causing dry and warm conditions over the $\text{WAM}_t$ target region. Thus we expect a causal relationship between these two regions. In summary, the first E0- cases in a given year seem to be influenced by both a warmer SAT and anomalous dry surface conditions over SAM. It seems plausible that warm SST anomalies over the SAT first cause a drying of the SAM region, which then propagates also into the $\text{WAM}_t$ region via a malfunctioning of moisture recycling.

## 4 Conclusions

Multiple studies have discussed the role of droughts and heat waves over the Amazon Basin. We focus on the role of atmo-spheric transport and show that two different transport mechanisms, linked to different ENSO conditions, are causing extreme drought under extreme heat over the western Amazon (Fig. 6). The transported air during E+ cases, occurring under El Niño conditions, is already warmer and drier 10 days prior to arrival over WAM. The air is originating from mid-tropospheric levels

over the tropical Atlantic Ocean, crossing the South American coast until it reaches the $\text{WAM}_t$ target region. This transport pat-tern highlights the importance of advection of dry and warm tropical air leading to extreme events over the Western Amazon. For E0- cases, occurring under neutral ENSO or La Niña conditions, we show that SAT and SAM are important source regions. These events are usually linked to pre-existing dry surface conditions over the southern Amazon and anomalously high SSTs over the southern Atlantic Ocean. However, most E0- cases show similar characteristics to the climatology until three days

prior to arrival, both in meteorological conditions as well as in mass transport, and the warm and dry anomalies seem to be caused only by the pre-existing warm and dry conditions over the continent. However, the E0- first events occurring in a season show already warmer and drier air 10 days prior to arrival. Therefore, the development of the first E0- extreme in a season is caused by advection of warm and dry air from the south Atlantic and further transport over the already dry southern Amazon, where evaporation is reduced. The subsequent events in a season are then less dependent on the conditions over the ocean and



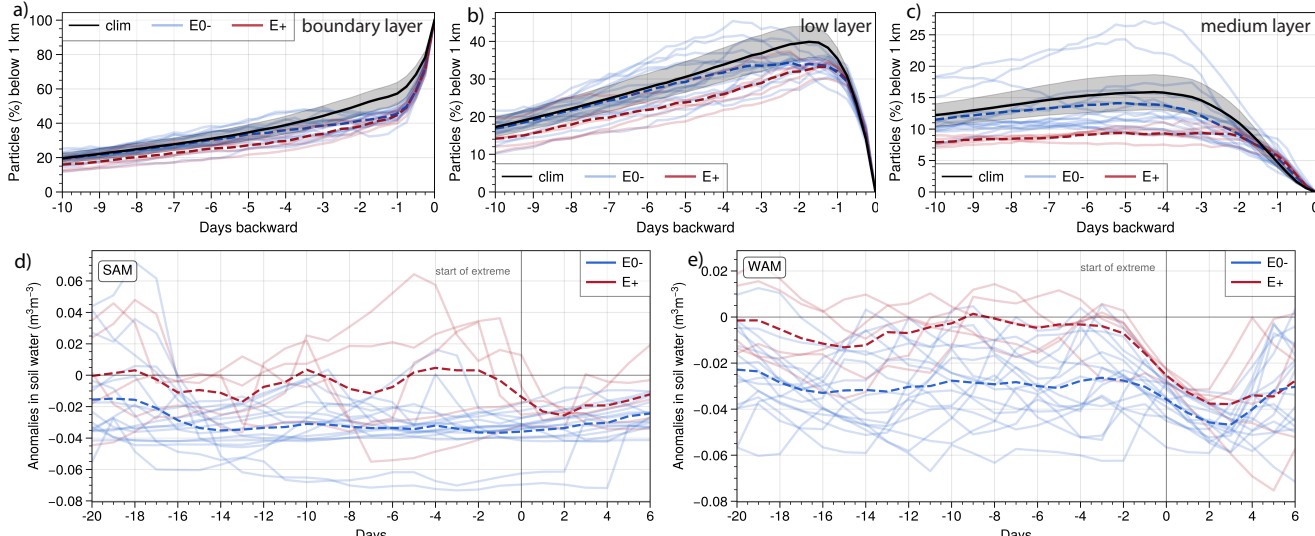

**Figure 5.** *Fraction of particles below 1 km and anomalies in volumetric soil water as a function of time.* (a, b, c) Fraction of particles below 1 km as a function of time for the particles arriving in the WAM$_t$ target region in the lowest layer (below 1 km) (a), low layer (1-2 km) (b) and medium layer (2-5 km) (c). Blue solid lines represent the E0- cases, red solid lines represent the E+ cases and the black solid line represents the climatological mean. Shaded grey areas represent one standard deviation around the climatological mean. (d, e) Anomalies in ERA5 volumetric soil water from 20 days before until 6 days after the start of each CDHE for the southern Amazon SAM (d) and the western Amazon WAM$_t$ (e).

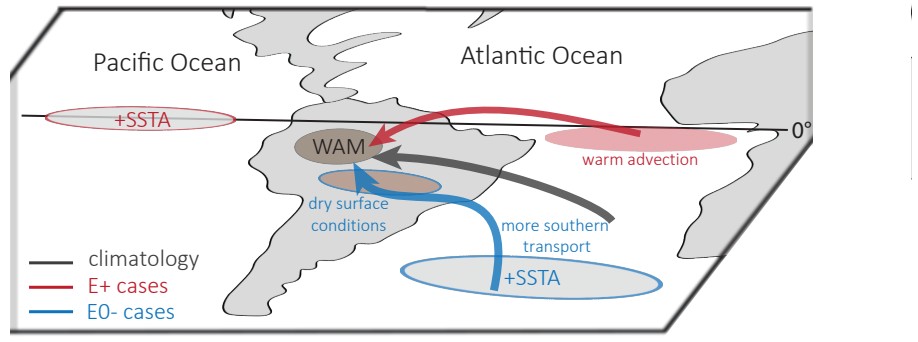

**Figure 6.** *Schematic illustration of the mass transport for climatology, E+ cases and E0- cases.*



are instead mainly influenced by a lack of moisture recycling over the pre-conditioned dry continental regions within the last few days of transport.

Our study adds an important perspective to the current research on extreme droughts and heat waves over the Amazon. It is motivated by the findings of Costa et al. (2022), who showed that heat waves over the southern Amazon are strongly linked to co-occurring dry surface conditions. Our study partly confirms this, as we show that for E0- cases the air is transported over the

anomalously dry southern Amazon, directly linking these two regions via atmospheric transport. Our study confirms previous results showing that reduction in moisture inflow is a main driver for extremes over the Amazon basin (Costa et al., 2022). Multiple studies have discussed the role of ENSO for the Amazon basin before (e.g. Jiménez-Muñoz et al., 2016; Marengo et al., 2008). Here, we have shown that different processes are causing CDHEs for E+ and E0- conditions.

Our results are consistent with previous studies with respect to the TAT and SAT being the major moisture sources for the

Amazon (e.g. Drumond et al., 2014; Leyba et al., 2023). While Drumond et al. (2014) only focus on the general moisture transport over the Amazon, our study also takes into account adiabatic and diabatic heating processes. Thus, we show that for E+ extremes warmer air from TAT is transported towards the western Amazon. Our results partly confirm that droughts over the Amazon are linked to positive SSTs over the Atlantic Ocean. Lewis et al. (2011) discuss that the droughts in 2005 and 2010 are associated with high Atlantic SSTs. Here we show that indeed the Atlantic Ocean SSTs play a crucial role for E0- cases.

However, no such relationship exists for E+ cases.

*Code and data availability.* The source code of FLEXPART is available from [https://gitlab.phaidra.org/flexpart/flexpart]. It can be used to fully reproduce the dataset, used in this study.

*Author contributions.* Katharina Baier led the study, wrote the paper and created the figures. Lucie Bakels and Andreas Stohl participated in

discussions and provided feedbacks on the text.

*Competing interests.* The authors declare that they have no conflict of interest.

*Acknowledgements.* This research was funded by the Austrian Science Fund (FWF) [10.55776/P34170]. For open access purposes, the author has applied a CC BY public copyright license to any author accepted manuscript version arising from this submission. ERA5 data were obtained using ECMWF's computing and archive facilities within a special project (spatvojt).




**Appendix A**

**Table A1.** *Extreme events over the western Amazon.* For each event the starting date of the extreme event is shown. The fraction of mass in %
is shown at x days prior arrival for the source regions; South Atlantic Ocean (SAT), southern Amazon (SAM), tropical Atlantic Ocean (TAT)
and whole Amazon region (AMZ). The average meteorological characteristic at 10 days prior arrival for the lowest vertical layer (0-1 km)
is shown for the height (z), potential temperature ($\theta$), temperature (T), latitude (lat) and specific humidity (q). The average boundary layer
height (m) for each extreme event is shown.

| DATE | SAT(-10d) | SAM(-4d) | TAT(-10d) | AMZ(-10d) | z(-10d) | $\theta$(-10d) | T(-10d) | Lat(-10d) | q(-10d) | blh(m) |
|---|---|---|---|---|---|---|---|---|---|---|
| 1997-09-17 | 10.7 | 0.1 | 66.6 | 4.3 | 3341 | 313.22 | 280.83 | -0.52 | 0.00401 | 968 |
| 2002-09-21 | 19.1 | 0.3 | 53.9 | 6.9 | 3206 | 312.48 | 281.26 | -2.98 | 0.00352 | 1201 |
| 2005-08-13 | 36.8 | 44.4 | 13.7 | 1.9 | 2646 | 305.35 | 280.59 | -14.70 | 0.00213 | 1442 |
| 2005-08-29 | 30.5 | 0.2 | 31.7 | 7.2 | 3012 | 310.79 | 280.47 | -2.83 | 0.00336 | 1228 |
| 2005-09-23 | 38.4 | 0.1 | 34.2 | 2.7 | 2382 | 309.49 | 284.39 | -8.66 | 0.00376 | 1044 |
| 2006-09-02 | 35.1 | 0.1 | 35.8 | 3.8 | 2603 | 309.58 | 284.46 | -2.74 | 0.00453 | 1407 |
| 2007-08-24 | 48.3 | 10.9 | 16.9 | 4.0 | 2458 | 306.35 | 281.03 | -12.03 | 0.00473 | 1305 |
| 2009-09-04 | 30.1 | 0.1 | 37.5 | 3.8 | 3239 | 312.16 | 280.82 | -3.44 | 0.00418 | 1223 |
| 2010-08-17 | 25.4 | 50.0 | 19.7 | 2.5 | 2925 | 311.19 | 281.48 | -10.11 | 0.00137 | 1350 |
| 2010-08-23 | 39.5 | 7.1 | 15.2 | 4.8 | 2446 | 305.49 | 278.53 | -14.28 | 0.00413 | 1127 |
| 2010-09-10 | 45.9 | 19.8 | 15.7 | 3.8 | 2578 | 308.17 | 282.06 | -11.99 | 0.00300 | 972 |
| 2011-08-08 | 30.9 | 6.0 | 33.9 | 1.8 | 3117 | 311.81 | 280.90 | -8.23 | 0.00247 | 1043 |
| 2011-08-26 | 40.0 | 10.0 | 27.6 | 2.4 | 2547 | 308.01 | 281.51 | -10.84 | 0.00305 | 948 |
| 2012-08-05 | 58.7 | 35.6 | 16.5 | 0.7 | 3893 | 316.58 | 278.52 | -15.90 | 0.00060 | 1075 |
| 2012-08-21 | 44.8 | 14.3 | 20.5 | 3.3 | 3678 | 310.93 | 275.78 | -13.55 | 0.00083 | 1173 |
| 2012-09-19 | 20.0 | 0.0 | 46.1 | 6.1 | 2950 | 311.61 | 282.67 | -4.16 | 0.00396 | 1129 |
| 2015-09-09 | 24.6 | 1.0 | 38.9 | 2.6 | 3350 | 313.62 | 280.70 | -2.06 | 0.00405 | 1108 |
| 2017-08-11 | 27.9 | 0.6 | 34.3 | 3.3 | 2421 | 308.63 | 284.51 | -6.30 | 0.00534 | 1134 |
| 2020-08-18 | 34.6 | 0.1 | 28.3 | 5.5 | 2662 | 309.53 | 282.67 | -2.63 | 0.00543 | 1437 |
| 2020-09-17 | 43.1 | 3.3 | 28.2 | 6.5 | 2560 | 309.93 | 283.94 | -7.62 | 0.00446 | 1046 |
| 2021-08-20 | 33.5 | 2.5 | 22.7 | 5.5 | 2141 | 306.09 | 283.87 | -9.60 | 0.00546 | 1069 |
| climatology | 32.6 | 9.4 | 25.2 | 2.8 | 2537 | 307.58 | 282.50 | -9.53 | 0.00417 | 1066 |




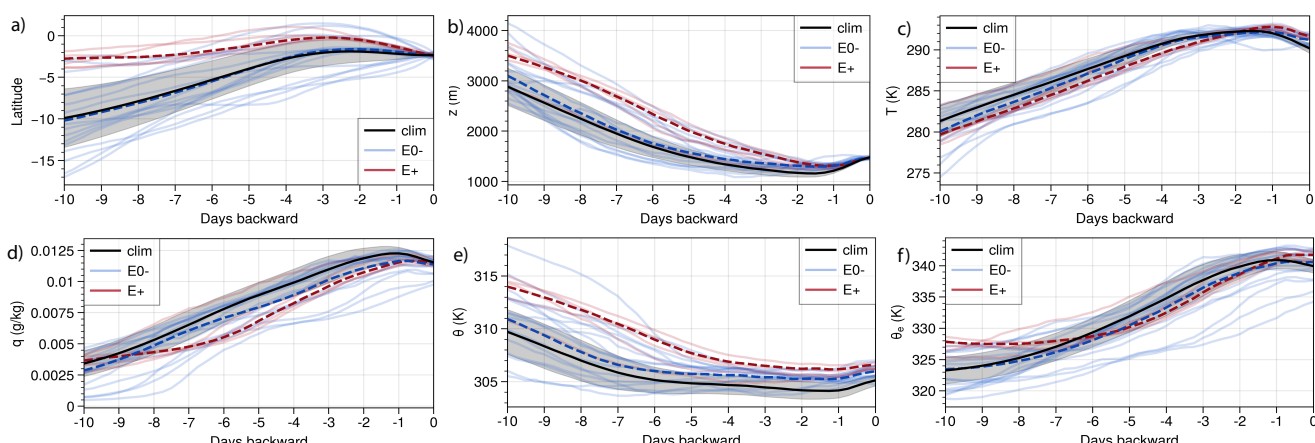

**Figure A1.** *Meteorological parameters as a function of time prior to arrival of particles starting in the low layer.* Same as in Figure 2, but for the low layer.

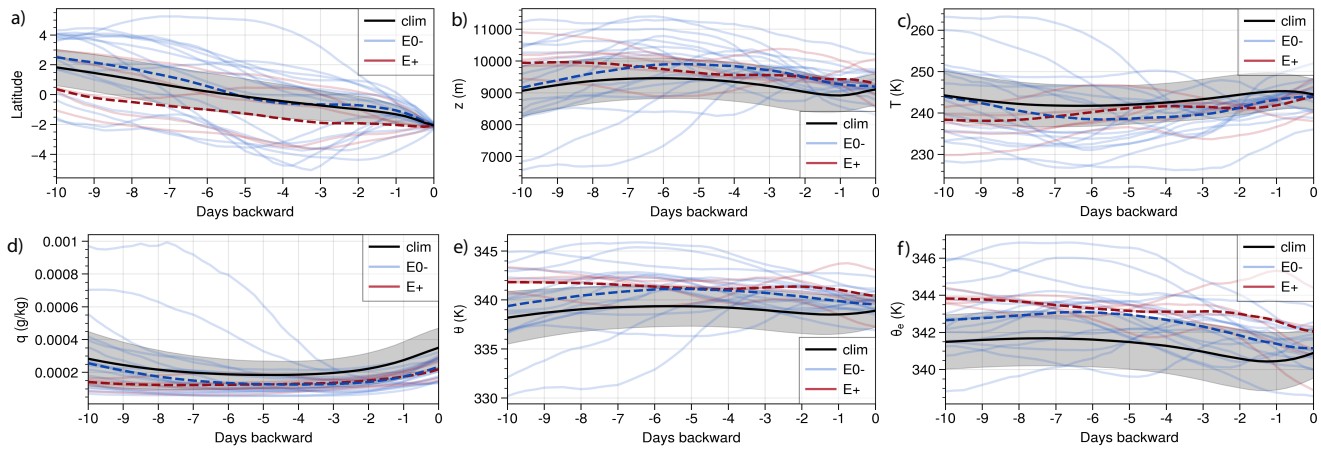

**Figure A2.** *Meteorological parameters as a function of time prior to arrival of particles starting in the high layer.* Same as in Figure 2, but for the high layer.



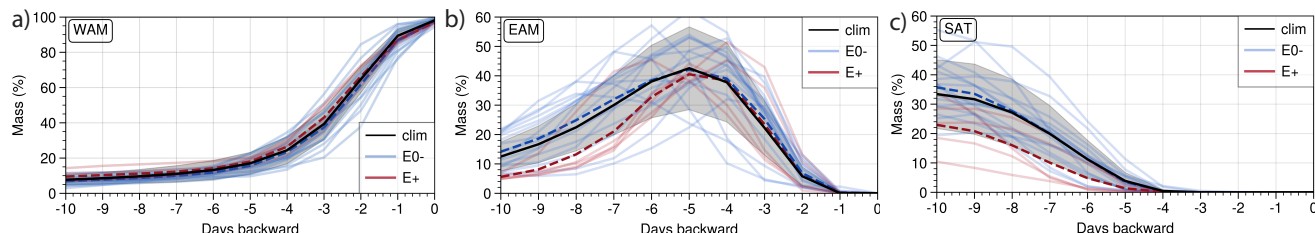

**Figure A3.** *Temporal evolution of mass transport in % for the following source regions: western Amazon WAM (a), eastern Amazon EAM (b), and South Atlantic Ocean SAT (c).*





**Figure A4.** *Mass transport anomalies for all CDHE events compared to the climatology, integrated over the whole atmospheric column and the 10-day tracking period.*



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
