# Peer review of "Two different mechanisms cause severe hot droughts in the western Amazon"

_EGUsphere, 2024_

## Referee Comment (RC2)

**General Comment**

The manuscript titled "Two different mechanisms cause severe hot droughts in the western Amazon" makes a significant scientific contribution regarding the influence of moisture and heat transport on drought events in the Amazon during neutral years and ENSO, utilizing a parcel tracking model. Below, I have outlined several comments that I suggest the authors consider to enhance the quality of the manuscript. Therefore, the manuscript has high scientific merit and discusses an important topic for both global and regional climate. Studies like this are crucial as it has been quantitatively challenging to assess transport (particularly of moisture) in South America. After addressing these comments, I recommend the work for publication.

I highlighted parts of the manuscript in sequence to make easy the identification.

**Main Comments**

1. The study focuses on neutral and ENSO events. In the conclusion, it makes some assessments regarding the Atlantic Ocean's sea surface temperature (SST) and its role in Amazonian droughts. However, understanding the practical relationship with the selected years is quite difficult, as there is no information on the SST anomalies in the Atlantic during these events, which could impact, as the authors themselves mention, the entire dynamics. I suggest including these assessments in the form of figures or graphs to support the presented results.

2. It is interesting that warm, dry air is transported to the Amazon region, especially in El Niño years. It would be valuable to add a discussion on whether this mass transport may inhibit a greater influx of water vapor from the forest, given that the work by Ruv Lemes et al. (2020)—referenced below—shows that greater moisture transport occurs internally within the Amazon during such events.

**Minor Comments**

1. The title should be revised for clarity regarding the manuscript's focus, as it currently lacks appeal.

2. The Amazon experiences different wet/dry periods; when referring to the "end of the dry season (August - September)," it is important to specify which region of the basin this refers to, or what reference is being used to indicate this.

3. Space between "laduse" = "land use".

4. Line 25, page 1 – Include and discuss the studies by Gatti et al., as they provide more current measurements to better describe the relationship between the Amazon and the carbon cycle.

Gatti, L. V., Basso, L. S., Miller, J. B., Gloor, M., Gatti Domingues, L., Cassol, H. L., ... & Neves, R. A. (2021). Amazonia as a carbon source linked to deforestation and climate change. *Nature*, 595(7867), 388-393.
Gatti, L. V., Cunha, C. L., Marani, L., Cassol, H. L., Messias, C. G., Arai, E., ... & Machado, G. B. (2023). Increased Amazon carbon emissions mainly from decline in law enforcement. *Nature*, 621(7978), 318-323.

5. Line 42, page 2 – It would be interesting to add a discussion on moisture transport in the Amazon. I suggest two articles that may assist. The first illustrates the differences in moisture transport during El Niño years, and the other explains the relationships arising from global warming and deforestation effects. Both could enhance the introduction and enrich it with characteristics of the regional climate.

Ruv Lemes, M. D. C., de Oliveira, G. S., Fisch, G., Tedeschi, R. G., & da Silva, J. P. R. (2020). Analysis of moisture transport from Amazonia to Southeastern Brazil during the austral summer. *Revista Brasileira de Geografia Física*, 13(06), 2650-2670.
Ruv Lemes, M., Sampaio, G., Fisch, G., Alves, L. M., Maksic, J., Guatura, M., & Shimizu, M. (2023). Impacts of atmospheric CO2 increase and Amazon deforestation on the regional climate: A water budget modelling study. *International Journal of Climatology*, 43(3), 1497-1513.

6. Line 45, page 2 – I suggest adding (and discussing) the following study:

Morales, J. S., Arias, P. A., Martinez, J. A., & Durán-Quesada, A. M. (2021). The role of low-level circulation on water vapour transport to central and northern South America: Insights from a 2D Lagrangian approach. *International Journal of Climatology*, 41, E2662-E2682.

7. I felt a lack of a more extensive explanation regarding the importance of the Amazon rainforest for moisture transport. Differences in pressure between the forest and the ocean, the role of vegetation, and flows in this mechanism could be better elucidated.

8. Again, why do the authors choose August and September as dry months and then include June and July? Was this based on a prior analysis or another study? I believe it is important to clarify this.

9. In results, line 113, page 4 – Personally, I find that acronyms of this magnitude (not widely known) disrupt the text's flow. I had to return several times to recall exactly what "CDHE" means—this is worth considering.

10. Line 128, page 5 – This was related to my comment 5. I think it is necessary to add literature and discuss the historical context of regional moisture transport dynamics. This is, by the way, an important part of the study.